# The Validity of Hawkin Dynamics Wireless Dual Force Plates for Measuring Countermovement Jump and Drop Jump Variables [note 1]

**DOI:** 10.3390/s23104820

**Published:** 2023-05-17

**Authors:** Andrew J. Badby, Peter D. Mundy, Paul Comfort, Jason P. Lake, John J. McMahon

**Affiliations:** 1Centre for Human Movement and Rehabilitation, University of Salford, Salford M6 6PU, UK; a.j.badby@edu.salford.ac.uk (A.J.B.); p.comfort@salford.ac.uk (P.C.); 2Hawkin Dynamics, Inc., Westbrook, ME 04092, USA; peter@hawkindynamics.com (P.D.M.); j.lake@chi.ac.uk (J.P.L.); 3School of Medical and Health Sciences, Edith Cowan University, Joondalup, WA 6027, Australia; 4Chichester Institute of Sport, University of Chichester, Chichester PO19 6PE, UK

**Keywords:** wireless dual force plate system, force–time, vertical jump, hardware, concurrent validity, agreement, accuracy

## Abstract

Force plate testing is becoming more commonplace in sport due to the advent of commercially available, portable, and affordable force plate systems (i.e., hardware and software). Following the validation of the Hawkin Dynamics Inc. (HD) proprietary software in recent literature, the aim of this study was to determine the concurrent validity of the HD wireless dual force plate hardware for assessing vertical jumps. During a single testing session, the HD force plates were placed directly atop two adjacent Advanced Mechanical Technology Inc. in-ground force plates (the “gold standard”) to simultaneously collect vertical ground reaction forces produced by 20 participants (27 ± 6 years, 85 ± 14 kg, 176.5 ± 9.23 cm) during the countermovement jump (CMJ) and drop jump (DJ) tests (1000 Hz). Agreement between force plate systems was determined via ordinary least products regression using bootstrapped 95% confidence intervals. No bias was present between the two force plate systems for any of the CMJ and DJ variables, except DJ peak braking force (proportional bias) and DJ peak braking power (fixed and proportional bias). The HD system may be considered a valid alternative to the industry gold standard for assessing vertical jumps because fixed or proportional bias was identified for none of the CMJ variables (n = 17) and only 2 out of 18 DJ variables.

## 1. Introduction

Lower body neuromuscular function (NMF) is most commonly evaluated during the performance of the countermovement jump (CMJ) and drop jump (DJ) tests [1,2], utilising force plates which enable the direct collection of force–time data [3,4]. Forward dynamics procedures are then used to calculate a multitude of variables from the resultant force–time curve, such as phase-specific velocity and displacement of the system’s centre of mass (which is equivalent to the body’s when no external load is being held) [5]. The introduction of commercially available, portable, and affordable wireless dual force plate systems (hardware and software) means there is a new opportunity for applied sport scientists, strength and conditioning coaches, and researchers to gain detailed information about their athletes’ NMF. This is important for researchers and practitioners who can now answer more authentic questions using wireless dual force plate data collected in practical real-world environments.

If force–time data are analysed instantly using bespoke proprietary software (i.e., those integrated into some force plate companies’ mobile applications), these data can be used immediately to inform athletes, coaches, and medical staff about individual preparedness and to recommended exercise prescriptions [5,6]. Although force plate testing is becoming commonplace in sport, accuracy must be maintained, as this is the main factor in determining an appropriate evaluation device [7]. Test results are only useful if the collection instruments (hardware and software) measure what they are supposed to (validity) [8]. Thus, the validity of any new technology should be established by quantifying the agreement between it and another well-established test device that is valid (i.e., a criterion “gold standard” device or method) [9,10,11,12,13].

The validity of a new technology established by quantifying its agreement with a “gold standard” device is considered a type of criterion-referenced validity, specifically concurrent validity [8,14]. For this purpose, a test of mean difference (e.g., a paired *t*-test) or a correlation test (e.g., Pearson’s correlation coefficient) between two test devices is considered insufficient [15], and it is recommended that agreement is established by calculating the limits of agreement (LOA) [9]. However, the outputs from LOA analyses do not account for the fact that fixed and proportional biases often interact [10,11,12,13,16,17,18]. An alternative approach to assess concurrent validity is through ordinary least products regression (OLPR), which provides a separate assessment of fixed and proportional bias as well as a prediction equation [10,16]. As such, OLPR is perhaps the only “philosophically” correct statistical technique to assess concurrent validity [16].

The validity of the Hawkin Dynamics, Inc. (HD) and ForceDecks (v2.0.7782) proprietary software has been appropriately investigated in previous research via OLPR analyses, in comparison to a criterion MATLAB script [6]. The authors reported a small magnitude of error for force–time variables from the CMJ, DJ, squat jump, and isometric mid-thigh pull tests using the HD software compared to a criterion MATLAB script [6]. This was attributed to similarities in analysis procedures between HD and MATLAB, such as phase identifications along the force–time curve and the utilisation of appropriate metric definitions and calculations [6].

One study to date, presented as a conference poster, aimed to establish the concurrent validity of the HD force plate hardware, but with limited statistical analyses (i.e., Pearson’s correlation coefficients and Bland–Altman plots, and not OLPR), and reporting of only CMJ outcome variables [19]. Similarly, researchers previously established the concurrent validity of portable 1-dimensional (i.e., collect vertical ground reaction forces only) [20] and 2-dimensional (i.e., collect vertical and horizontal ground reaction forces) [21] PASCO force plate options against a criterion device (i.e., a fixed in-ground Kistler force plate system), but did not include the assessment of bias using appropriate statistics (i.e., OLPR statistics) [20,21]. To establish system accuracy, determining any systematic disagreement between said apparatus and a widely used and thoroughly investigated “gold standard” system using appropriate agreement statistics is critical [10,18]. Due to the validity of the HD software having been established [6], an assessment of the validity of the HD force plate hardware using appropriate agreement statistics is required to provide confidence to users regarding the accuracy of the hardware.

Therefore, the purpose of this study was to determine the concurrent validity of the HD force plate hardware by assessing the agreement between selected outcome (e.g., jump height (JH), flight time, etc.) and strategy (e.g., time to take-off, ground contact time (GCT), etc.) variables during the CMJ and DJ tasks, compared to those derived from a laboratory-grade, in-ground force plate system (i.e., a “gold standard”). It was hypothesised that agreement would be found between the force plate systems for all CMJ and DJ variables with no fixed or proportional bias present. The results of this study will inform the use of the HD force plate system in future research projects and in applied sports settings where system accuracy is paramount.

## 2. Materials and Methods

### 2.1. Participants

Twenty recreationally active adults (age = 27 ± 6 years, body mass = 85 ± 14 kg, height 176.5 ± 9.23 cm) with varied sports backgrounds (e.g., amateur soccer, netball, weightlifting, etc.) and who were free from any injury that would prevent them performing maximum effort trials volunteered to participate in the study. Current training status and previous resistance and vertical jump training experience were not a limiting factor in this study, due to its focus on agreement between the two force plate systems alone. Informed consent was provided, and the study was pre-approved by the Institutional Ethics Committee (application ID 2768) before recruitment and testing commenced.

### 2.2. Design

A cross-sectional design was employed, whereby testing was conducted during a single session in the human performance laboratory at the University of Salford on 17 December 2021. A standardised warm-up (~10 min) consisting of dynamic stretching and submaximal CMJs and DJs was performed by each participant prior to testing to reduce the risk of injury.

### 2.3. Force Plate Setup

An HD force plate system (3rd Generation, model 0484; Westbrook, Maine, USA) consisting of two portable adjacent force plates was placed directly on top of and within the dimensions of two adjacent Advanced Mechanical Technology, Inc. (AMTI; Model Biomechanics Measurement Series 400600; Watertown, MA, USA) in-ground force plates to collect the ground reaction force (GRF) produced through each leg independently and simultaneously at 1000 Hz (Figure 1 and Figure 2). The GRF data were sampled for five (CMJ) and six (DJ) seconds using HD proprietary software and Qualisys Track Manager software (Qualisys Ltd., Gothenburg, Sweden) for the HD and AMTI systems, respectively. Both systems were zeroed before each trial so that the weight of the HD system was removed from the AMTI system before data acquisition. The raw (i.e., unfiltered) vertical component of the GRF (vGRF) data series was exported from each force plate system’s software to Microsoft Excel, which was used to analyse the bilateral forces (summed left and right leg forces) using a custom spreadsheet. The vGRF data exported from both force plate systems were analysed identically using the custom spreadsheet.

### 2.4. Countermovement Jump

For the CMJ trials, participants stepped onto the force plates and stood completely upright (extended hips and knees) and motionless for at least one second before completing three maximal effort trials following a “3, 2, 1, jump” command. Participants were cued to jump “as fast (i.e., strategy) and high (i.e., outcome) as possible” with arms akimbo. Standardising performance with this cue was done to encourage the participants to perform trials with the “intent” of attaining the maximal height achievable, but with a focus on performing a fast movement throughout the unweighting, braking, and propulsive phases (i.e., a short time to take-off) to prevent them favouring a fast or high strategy independently [22].

Each participant’s body weight was calculated by averaging the vertical force trace over the first one second of data collection when the subject was stationary on the force plates [23,24,25]. The onset of movement in the CMJ was defined as the 30 ms before the instant that the vGRF exceeded the mean plus or minus five times the standard deviation (SD) of the average force calculated during the weighing phase [24]. This was also when numerical integration began [24]. The net impulse was calculated by numerically integrating the net force–time record and summing all of the individual net impulses on a sample-by-sample basis [20]. The vertical velocity of the centre of mass (COM) was determined on a sample-by-sample basis by dividing vGRF (minus body weight) by system mass, and then integrating the product using the trapezoid rule [20,23]. Then, the vertical displacement of the COM was determined on a sample-by-sample basis by integrating the velocity–time record using the trapezoid rule [23,26]. The power applied to the COM was determined by multiplying force by velocity on a sample-by-sample basis [24].

Take-off and touchdown force thresholds for the CMJ were determined as equal to 5 SDs of the vGRF during the first 300 ms of flight [23]. Due to the AMTI system demonstrating greater flight-phase force, the AMTI system’s take-off and touchdown thresholds were used for both systems. For comparison, 5 SDs of the vGRF during the first 300 ms of the flight phase equalled ~16 N vs. ~8 N for the AMTI and HD systems, respectively; thus, 16 N was applied to both systems. The 16 N take-off threshold was used to time-align the CMJ data between the AMTI and HD systems. This was necessary as the AMTI data capture was triggered first, followed immediately by HD data capture. Thus, there were more AMTI data prior to take-off and so the difference in vGRF samples between systems from the start of capture to take-off was deducted from the AMTI data (Figure 3).

The CMJ phases were identified using methods explained and used recently [5]. The braking phase was defined as the period between the instant of peak negative velocity and the instant of zero velocity [27]. The propulsive phase was defined as the period between the velocity exceeding 0.01 m/s and the instant of take-off [27]. The CMJ variables included in the analyses and a description of how they were calculated are shown in Table 1.

### 2.5. Drop Jump

For the DJ trials, participants stepped onto a 45 cm box placed 2 cm behind the force plate system (Figure 1 and Figure 2). However, because the HD force plates had a height of 6.5 cm, but were positioned on top of the AMTI force plates 1.5 cm below the rubber surface the box was placed on, the distance from the box to the top surface of the force plates (i.e., effective box height) was 40 cm. This was arranged because 40 cm is a commonly used box height for DJ assessments, following proposed methods for evaluating DJ performance using only one on-ground force plate system [28]. Following a “3, 2, 1, drop” command, participants dropped (without consciously stepping down or jumping upwards) from the box onto the force plates and performed a maximal effort DJ with a focus on executing the jump with as short a contact time as possible whilst also aiming to maximise JH [22]. Following landing, participants stood completely upright (extended hips and knees) and motionless for at least one second [28]. Each participant’s body weight was calculated by averaging the vertical force trace over the final one second of data collection when the subject was stationary on the force plates after landing [28].

The DJ vGRF data were time aligned between force plate systems by commencing data analysis (i.e., phase identification and numerical integration) from the first sample vGRF that surpassed a touchdown threshold of 25 N (Figure 4). Thus, no vGRF data from the AMTI system (which began data capture immediately before the HD system) that were recorded before touchdown were included in the analyses. A 25 N threshold was also used to identify take-off and the second touchdown (post DJ). The numerical integration of the vGRF–time record and identification of the braking and propulsive phases were conducted as per the CMJ data except that they began at the instant of touchdown. The touchdown velocity, which was the velocity value used to commence numerical integration, was estimated using a recently validated method [28]. Fall height was then estimated as touchdown velocity squared divided by 2 × 9.81 [29]. The DJ variables included in the analyses and a description of how they were calculated are shown in Table 1.

### 2.6. Statistical Analysis

The average across three CMJ and DJ trials (for each variable) was taken forward for statistical analysis. Statistical analyses were performed using SPSS software (version 25; SPSS Inc., Chicago, IL, USA) using nonlinear regression and the user-defined loss function [10]. The agreement between force plate systems was determined via OLPR, which was conducted following recommendations in the literature [10,18]. If the bootstrapped 95% confidence interval (CI) for the intercept did not include 0, fixed bias was inferred to be present. If the bootstrapped 95% CI for the slope did not include 1, proportional bias was inferred to be present.

## 3. Results

Descriptive statistics, OLPR coefficients, and corresponding bootstrapped 95% CIs are reported in Table 2 (CMJ) and Table 3 (DJ). For all CMJ variables investigated, there was no fixed or proportional bias between the two force plate systems. The same was seen for DJ variables except that both fixed and proportional bias were identified for peak braking power and proportional bias was identified for peak braking force.

Descriptive statistics (mean ± standard deviation) for fall height and touchdown velocity for the AMTI and HD force plate systems are reported in Table 4. The mean fall height recorded on the AMTI and HD systems was approximately 5 cm less than the prescribed 40 cm effective box height (Table 4).

## 4. Discussion

The purpose of this study was to determine the concurrent validity of a wireless portable dual force plate system by assessing the agreement between selected force–time variables collected during the CMJ and DJ tests using the test system and those collected using an in-ground AMTI system, considered a “gold standard”. This was carried out because the validity of the HD proprietary software was established using the CMJ and DJ tests in previous research [6], and previous attempts to validate the HD hardware using only the CMJ test were performed with a lack of methodological and statistical distinction [19]. Based on the results of this study, the wireless dual force plate system can be considered a valid alternative to the criterion industry gold standard with respect to collecting CMJ and DJ force–time data, because the OLPR analysis showed no fixed or proportional bias between the two force plate systems for any of the CMJ variables (n = 17; Table 2), and bias was shown for only 2 out of the 18 DJ variables (Table 3).

### 4.1. Agreement Considerations

These findings support the conclusions of the sole previous study conducted with a similar approach [19]; however, the results here indicate a better agreement between the two force plate systems. This is likely due to this study having applied what may be considered a philosophically more robust methodological and statistical design approach. For example, in the study by Crowder et al. [19], it is unclear whether their participants performed three maximal effort CMJs separately on the HD and AMTI systems. This is an initial concern, as it is rare that participants will perform separate CMJ trials with identical force–time characteristics [30]. This introduces random error due to inherent biological variation, which confounds the mechanical variation under investigation. Additionally, the study by Crowder et al. [19] assessed the mean bias between systems for the CMJ outcome measure JH alone, without assessing agreement between the strategy metrics underpinning JH. In contrast, the present study performed a more thorough analysis by including both outcome and strategy variables in the CMJ test.

From a statistical perspective, the more robust OLPR analysis was chosen in this study, according to recommendations in previous literature [10,18], as opposed to the Pearson’s correlation coefficients and Bland–Altman plots with 95% LOA used by Crowder et al. [19]. Differences in data collection frequencies were also evident, with the AMTI system collecting at 1200 Hz, whereas the HD system collected at 1000 Hz [19]. This discrepancy can affect key events of the CMJ, such as onset of movement and take-off thresholds, although it is unknown by how much.

Additionally, Crowder et al. [19] allowed participants to use arm swing during trials, which adds another factor which could have affected the variability of trials if data were not collected simultaneously on both force plates. The authors highlighted that the inclusion of arm swing could have increased the variability of the trials, as has been seen in previous research [19]. Taken together, the methodological shortcomings of the Crowder et al. [19] study may explain why their LOA analysis showed that the average JH collected across three trials with the HD system could be expected to range from 7.10 cm lower to 7.63 cm higher than that measured by the AMTI system, which the authors of this study deem unacceptable.

Furthermore, the CMJ is used within a testing battery specifically as a measure of slow SSC capacity, and generally demonstrates lower rates (i.e., increased movement time) and magnitudes (i.e., lower peak braking and propulsive force) for specific force–time measures in comparison to tests of fast SSC capacity (i.e., the DJ test), when performed with the correct technique (i.e., cued to jump “as fast and high as possible”, with a DJ GCT of <250 ms) (Table 1 and Table 2). In the present study, we performed a more thorough analysis by also including outcome and strategy variables for the DJ test. These analyses may have been justified by the results of this study, which identified fixed and proportional bias in peak braking power and proportional bias in peak braking force for the DJ (Table 2), but not the CMJ (Table 1).

Peak braking force is a measure that has been of interest to research related to identifying injury risk mechanisms of the lower extremities (e.g., anterior cruciate ligament injuries) during drop landing and DJ tasks [31], and can be influenced by the strategy applied (i.e., a “stiffer” strategy with less range of motion through braking would increase peak braking force) [22,30]. A potential reasoning for the proportional bias in peak braking force in DJs is a difference in the resolution and accuracy (i.e., the precision) of the load cells used in the hardware. The HD hardware utilises four strain-gauge-based “beam” load cells per plate (one at each corner of the force plate) with a reported rounding resolution and accuracy of ±0.25 N and 0.1 N, respectively [32].

In contrast, laboratory-grade, in-ground force plate systems (i.e., the “gold standard”) might use more expensive sensors with better resolution and accuracy (e.g., to ± 0.1 N per sample), but these usually come at a greater cost to the consumer. In this example, a sample of force–time data representing 0.8 N of vGRF would round lower to 0.75 N if using the HD hardware but would remain as 0.8 N if using the “gold standard” (e.g., the AMTI) hardware. Thus, as peak braking force increases, there is the potential for a greater accumulated difference in rounding per sample in the HD system which may have caused this discrepancy. Additionally, the raw vGRF was analysed in this study but filtering the force–time record may have improved the agreement in peak braking force. The proposed rounding errors which caused bias to be present in the peak braking force also extended to the peak braking power, because power was determined by multiplying force by velocity on a sample-by-sample basis.

Despite these findings, peak braking force and power only represent a single instant in time and may not contribute significantly to the outcome (i.e., they do not significantly affect the net impulse production or TOV) of vertical jump tasks. Moreover, based on additional calculations using linear regression (i.e., y = mx − c, as produced by plotting the grand mean of the two force plate systems against the percentage difference between the two force plate systems), the predicted percentage difference between force plate systems at 8000 N of DJ peak braking force (~40% larger than the mean grand mean reported in this study) is only 3.6%. Similarly, the predicted percentage difference between force plate systems for 12000 W of DJ peak braking power (~40% larger than the grand mean for this study) is only 5.7%. As such, the proportional bias demonstrated in peak braking force and peak braking power in the DJ test is relatively small, even at extreme predicted values for each metric, and likely falls within the expected standard error of measurement.

These findings are reasonable considering that the force plate systems produced by AMTI are more expensive stationary systems manufactured towards a clinical, laboratory-style market, but, in contrast, the HD system provides greater practicality at a lower price point, with its target consumers being practitioners working in the field. A change to more resolute and accurate sensors like those used in the AMTI system would result in large increases in the cost of production which would reduce the HD system’s affordability for practitioners. However, despite the target consumers of the HD product being practitioners working in the field, the agreement demonstrated in this study illustrates an ability for the HD force plate system to also be a cheaper, more practical, and valid alternative in the clinical, laboratory-style force plate market.

### 4.2. Drop Jump Considerations

A proposed limitation to the DJ assessment is that previous research has reported differences between effective box height and actual fall height [33]. A difference in fall height would change the touchdown velocity and thus the force–time characteristics of the braking phase (e.g., a change in net braking impulse), and, if using the DJ as a task during training, it may affect the accuracy of the prescription of DJ training load. Using a sample of twenty-two physical education students, Geraldo et al. [33] observed a progressively increasing difference (an average of 2 cm increase in difference per 10 cm increment in effective box height) between effective box height and fall height at effective box heights of 20, 30, 40, and 50 cm using an AMTI force plate sampling at 1000 Hz, similar to the methods used in this study.

From as low as a 20 cm effective box height, Geraldo et al. [33] reported a fall height of 13.7 ± 1.6 cm which equated to an average difference of 6.3 cm. Discrepancies of this amount have also been reported in research with senior, elite, male rugby players, in which poor relative reliability was observed for average fall height which ranged from 14.87 to 29.85 cm [34]. Geraldo et al. [33] also reported a fall height of 29.4 ± 2.6 cm from an effective box height of 40 cm, which demonstrated an average difference of 10.6 cm. The results of the present study corroborate these findings, reporting a fall height of 0.35 ± 0.04 m from the same effective box height, equating to a lower average difference of 5 cm, which was the same for both the AMTI and HD systems (Table 2).

These results indicate that participants stepped down from the box by an average of 5 cm from the effective box height during each DJ trial. Therefore, caution is advised when comparing DJ performance between individuals or results from separate studies due to potential discrepancies between fall height and effective box height, as identified in this study (Table 2), and the variation in the amount of difference reported in previous research [33,34]. Inaccuracies in the calculated TDV will result in an incorrect calculation of force–time variables throughout the remainder of the task, such as braking net impulse. The study by Merrigan et al. [6] compared DJ data collected on a single portable Bertec Corp. force platform when analysed in MATLAB vs. HD proprietary software; however, the TDV was predicted based on box height (based on the work–energy theorem), which was assumed to be the same for every participant tested. Consequently, it can be suggested that the validation of some DJ force–time variables between software tools performed in this study is inaccurate due to inaccuracies in fall height and TDV [6]. Future research should continue to establish fall height if using the DJ test as an assessment tool due to the differences reported in this study between effective box height and fall height.

The results of this study confirm agreement of the HD force plate hardware with the gold standard for common CMJ and (most) DJ force–time measures. Additionally, previous research validated the HD proprietary software using the same assessments [6]. The bias in specific DJ variables identified in this study was attributed to load cell resolution and rounding, but the mean differences were not deemed to be meaningful. Taken together, in addition to the discrepancies between DJ effective box height and fall height identified in this study, future research should consider establishing the concurrent validity of both the hardware and software combined against a criterion hardware and software system whilst using tests (e.g., the 10/5 and countermovement rebound jump tests) which produce different magnitudes, rates, and frequencies of loading (i.e., vGRF production) and eradicate the issues of fall height discrepancies. Additionally, the strain-gauge “beam” load cell sensors of the HD force plates should be compared to other portable force plate systems including those that use piezoelectric sensors, as was performed in the previous research examining the concurrent validity of portable PASCO force plates (strain-gauge sensors) against criterion in-ground Kistler force plates (piezoelectric sensors) [20,21].

### 4.3. Practical Considerations

Based on the present study’s results, the HD system can be considered an accurate system to use in any setting, and as an alternative to traditional, nonportable, and more expensive in-ground force plate systems for the CMJ test. This consideration also extends to the DJ test except for peak braking force and peak braking power. Although these measures may be of interest in specific settings (e.g., for monitoring DJ peak braking force during injury rehabilitation), these measures represent only an instant in time and had no effect on the agreement of any of the other outcome or strategy variables assessed. Additionally, the predicted percentage differences for these metrics between systems at 40% above the presented grand mean were considered small and within the expected measurement error. These results are useful for practitioners who are seeking a force plate system to evaluate NMF using CMJs and DJs in sports, but are restricted by system complexity, location, and price.

Researchers and practitioners should be mindful that the data presented here are related to comparisons between two specific force plate systems (i.e., HD and AMTI) which both utilise strain-gauge-based beam load cells. The findings here cannot be extrapolated to provide a rationale to use the HD system over other force plate systems such as those which use piezoelectric load cells, for which agreement is yet to be determined. Additionally, the sample size is acknowledged as a limitation of the study. Whilst we acknowledge this, other studies that have assessed the concurrent validity of PASCO portable force plates for measuring CMJ variables used only 2 participants [21], albeit with several repetitions performed per participant, and a similar study by Lake et al. [20] utilised 28 participants.

## 5. Conclusions

The results of this study demonstrate that there is no fixed or proportional bias between the HD and AMTI (gold standard) force plate systems for measuring common CMJ strategy and outcome variables. Therefore, the HD force plate system may be considered a valid alternative to the industry gold standard for the assessment of CMJ force–time characteristics and thus may be confidently used for this purpose by researchers and practitioners alike.

The HD system is also a valid alternative to an industry gold standard force plate system for measuring common DJ force–time characteristics, because although bias was present for peak braking force and peak braking power, the predicted percentage differences at high metric values (8000 N and 12,000 W, respectively) were small and had no impact on any other outcome or strategy variables that are likely to be of interest to researchers and practitioners conducting DJ testing. Although accuracy has been established for the HD force plate system, it is still advised to utilise the same force plate system for each testing occasion (i.e., do not use the HD and AMTI systems interchangeably when testing athletes at different time points).

## Figures and Tables

**Figure 1 sensors-23-04820-f001:**
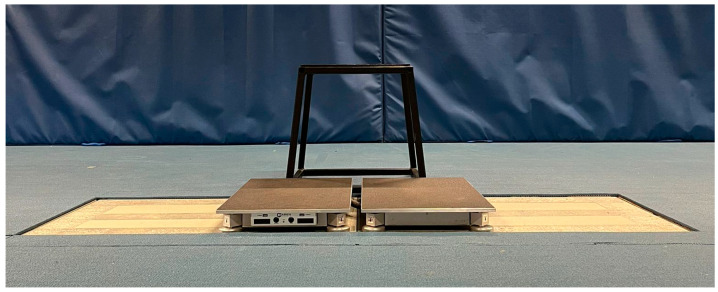
Example set-up for data collection (frontal plane).

**Figure 2 sensors-23-04820-f002:**
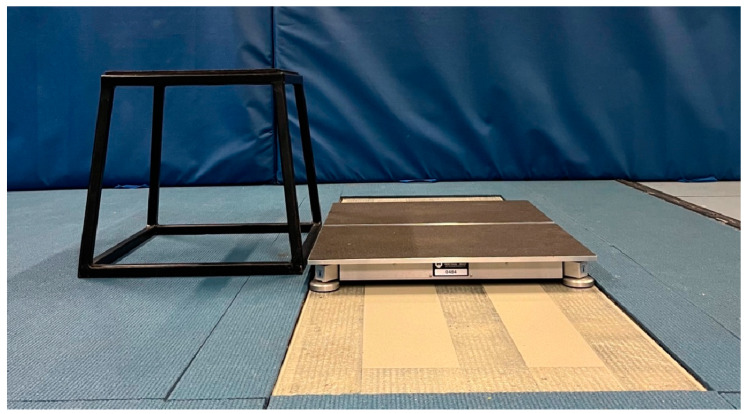
Example set-up for data collection (sagittal plane).

**Figure 3 sensors-23-04820-f003:**
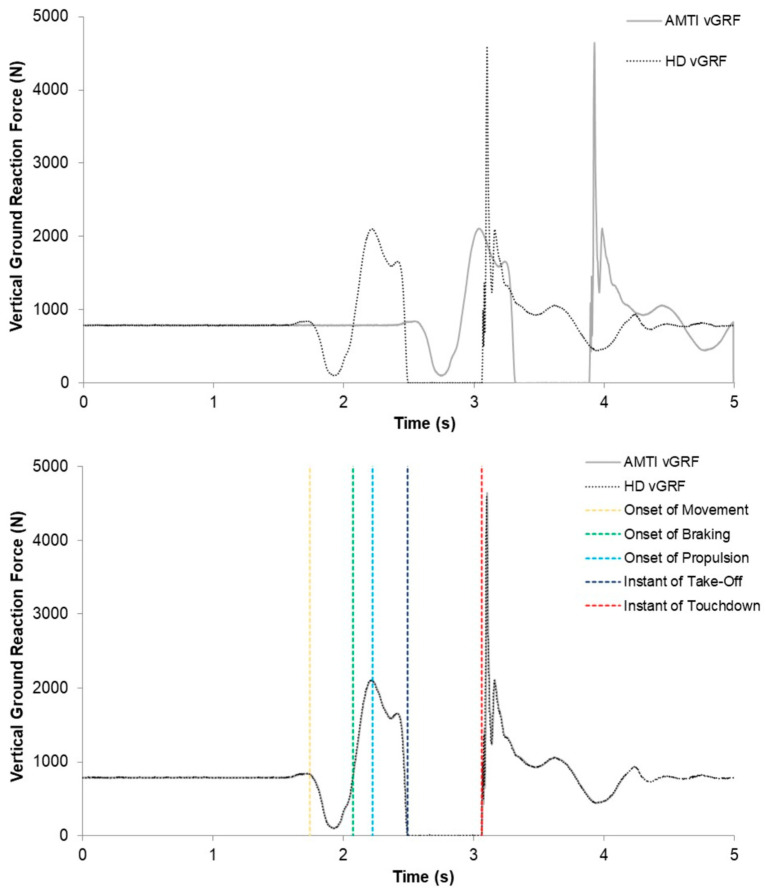
A representative example of an original (**top**) and time-aligned (**bottom**) countermovement jump trial recorded by the AMTI (solid grey line) and HD (dotted black line) force plate systems. The bottom graph also illustrates the occurrence of key events. vGRF = vertical ground reaction force.

**Figure 4 sensors-23-04820-f004:**
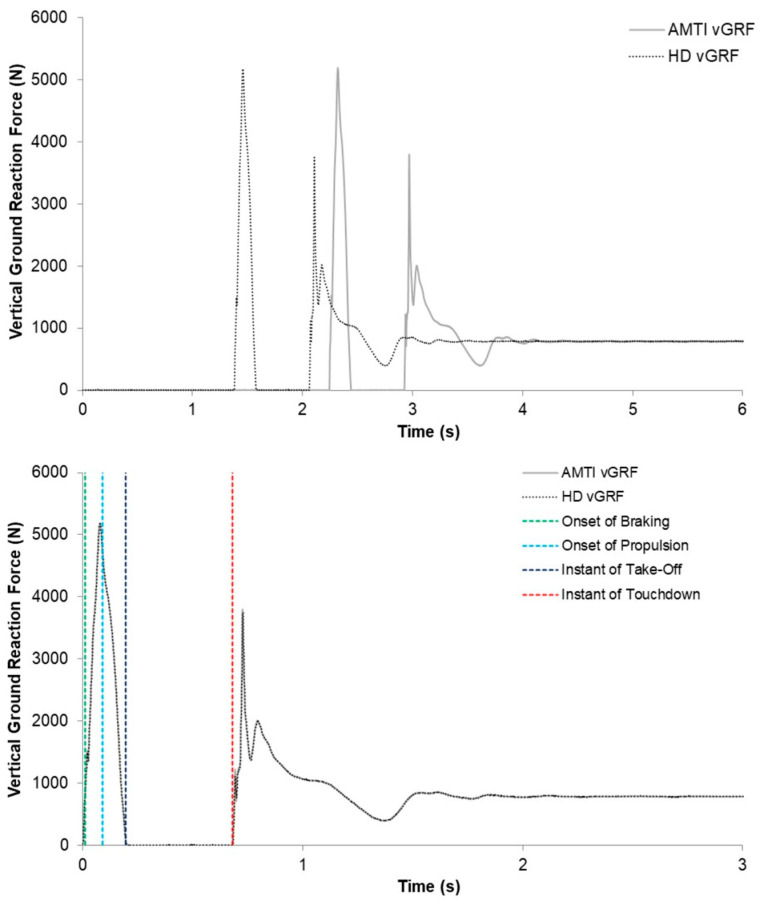
A representative example of an original (**top**) and time-aligned (**bottom**) drop jump trial recorded by the AMTI (solid grey line) and HD (dotted black line) force plate systems. The bottom graph also illustrates the occurrence of key events. vGRF = vertical ground reaction force.

**Table 1 sensors-23-04820-t001:** Selected force–time variable calculations.

Variables	Calculation	CMJ?	DJ?
**RSI (AU)**	Jump Height divided by Ground Contact Time.		X
**mRSI (AU)**	Jump Height divided by Time to Takeoff.	X	
**Jump Height (m)**	The change in centre of mass position between the instant of take-off and peak positive vertical displacement of the centre of mass during the flight phase, calculated as takeoff velocity squared divided by 19.62.	X	X
**Flight Time (s)**	The time taken to complete the flight phase.	X	X
**Ground Contact** **Time (s)**	The total time taken from instant of touchdown the instant of take-off.		X
**Time to** **Takeoff (s)**	The total time taken from the onset of movement to the instant of take-off.	X	
**Mean Propulsive** **Power (W)**	The mean mechanical power applied to the centre of mass during the propulsive phase.	X	X
**Peak Propulsive** **Power (W)**	The peak instantaneous mechanical power applied to the centre of mass during the propulsive phase.	X	X
**Peak Propulsive** **Velocity (m/s)**	The peak instantaneous vertical velocity of the centre of mass during the propulsive phase.	X	X
**Net Propulsive** **Impulse (N.s)**	The net vertical impulse applied to the centre of mass during the propulsive phase.	X	X
**Mean Propulsive** **Force (N)**	The mean vertical ground reaction force applied to the centre of mass during the propulsive phase.	X	X
**Peak Propulsive** **Force (N)**	The peak instantaneous vertical ground reaction force applied to the centre of mass during the propulsive phase.	X	X
**Stiffness (N/m)**	Peak braking force divided by braking depth.		X
**Braking Depth (m)**	The peak negative vertical displacement of the centre of mass during the braking phase.		X
**Countermovement** **Depth (m)**	The peak negative vertical displacement of the centre of mass during the braking phase.	X	
**Mean Braking** **Power (W)**	The mean mechanical power applied to the centre of mass during the braking phase.	X	X
**Peak Braking** **Power (W)**	The peak negative instantaneous mechanical power applied to the centre of mass during the braking phase.	X	X
**Net Braking** **Impulse (N.s)**	The net vertical impulse applied to the centre of mass during the braking phase.	X	X
**Mean Braking** **Force (N)**	The mean vertical ground reaction force applied to the centre of mass during the braking phase.	X	X
**Peak Braking** **Force (N)**	The peak instantaneous vertical ground reaction force applied to the centre of mass during the braking phase.	X	X

**Key:** RSI, reactive strength index; mRSI, modified reactive strength index; AU, arbitrary unit; m, metres; s, seconds; W, watts; N, Newtons; CMJ, countermovement jump; DJ, drop jump; X, included.

**Table 2 sensors-23-04820-t002:** Descriptive and agreement statistics for the selected CMJ variables.

Variables	AMTI			HD			Intercept			Slope		
	(Mean ± SD)			(Mean ± SD)			* 95% CI *			* 95% CI *		
**mRSI (AU)**	0.43	±	0.10	0.43	±	0.10	0.00			1.01		
							* −0.01 *	* to *	* 0.00 *	* 0.99 *	* to *	* 1.04 *
**Jump Height (m)**	0.31	±	0.07	0.31	±	0.06	−0.01			1.03		
							* −0.02 *	* to *	* 0.01 *	* 0.98 *	* to *	* 1.08 *
**Flight Time (s)**	0.51	±	0.05	0.51	±	0.05	0.00			1.00		
							* −0.01 *	* to *	* 0.01 *	* 0.99 *	* to *	* 1.01 *
**Time to** **Takeoff (s)**	0.76	±	0.09	0.77	±	0.09	−0.01			1.01		
							* −0.06 *	* to *	* 0.00 *	* 0.96 *	* to *	* 1.04 *
**Mean Propulsive** **Power (W)**	2316.23	±	481.89	2302.91	±	480.00	4.24			1.00		
							* −33.80 *	* to *	* 42.29 *	* 0.99 *	* to *	* 1.02 *
**Peak Propulsive** **Power (W)**	4124.56	±	907.17	4107.07	±	913.99	48.12			0.99		
							* −17.08 *	* to *	* 113.32 *	* 0.98 *	* to *	* 1.01 *
**Peak Propulsive** **Velocity (m/s)**	2.59	±	0.25	2.58	±	0.24	−0.07			1.03		
							* −0.19 *	* to *	* 0.04 *	* 0.99 *	* to *	* 1.08 *
**Net Propulsive** **Impulse (N.s)**	210.34	±	41.74	209.25	±	42.08	2.79			0.99		
							* −1.04 *	* to *	* 6.62 *	* 0.97 *	* to *	* 1.01 *
**Mean Propulsive** **Force (N)**	1667.56	±	291.81	1664.07	±	290.81	−2.26			1.00		
							* −12.43 *	* to *	* 7.91 *	* 1.00 *	* to *	* 1.01 *
**Peak Propulsive** **Force (N)**	2042.72	±	343.63	2040.87	±	343.97	3.92			1.00		
							* −5.25 *	* to *	* 13.09 *	* 1.00 *	* to *	* 1.00 *
**Countermovement** **Depth (m)**	−0.30	±	0.06	−0.30	±	0.06	0.01			1.01		
							* −0.01 *	* to *	* 0.02 *	* 0.97 *	* to *	* 1.06 *
**Mean Braking** **Power (W)**	−1092.83	±	274.27	−1097.65	±	273.88	6.37			1.00		
							* −2.11 *	* to *	* 14.85 *	* 0.99 *	* to *	* 1.01 *
**Peak Braking** **Power (W)**	−1518.54	±	426.54	−1524.72	±	426.16	7.53			1.00		
							* −0.77 *	* to *	* 15.83 *	* 1.00 *	* to *	* 1.01 *
**Net Braking** **Impulse (N.s)**	106.79	±	25.06	107.32	±	24.97	−0.91			1.00		
							* −2.07 *	* to *	* 0.26 *	* 0.99 *	* to *	* 1.02 *
**Mean Braking** **Force (N)**	1495.80	±	250.73	1498.44	±	250.90	−1.58			1.00		
							* −10.66 *	* to *	* 7.49 *	* 0.99 *	* to *	* 1.01 *
**Peak Braking** **Force (N)**	1951.83	±	319.75	1952.70	±	320.31	2.53			1.00		
							* −11.80 *	* to *	* 16.86 *	* 0.99 *	* to *	* 1.01 *
**Body Weight (N)**	833.21	±	141.92	833.69	±	141.68	−1.88			1.00		
							* −7.80 *	* to *	* 4.03 *	* 1.00 *	* to *	* 1.01 *

**Key:** AMTI, advanced mechanical technology, Inc.; HD, Hawkin Dynamics; SD, standard deviation; CI, confidence interval; AU, arbitrary unit; m, metres; s, seconds; N, Newtons.

**Table 3 sensors-23-04820-t003:** Descriptive and agreement statistics for the selected DJ variables.

Variables	AMTI			HD			Intercept			Slope		
	(Mean ± SD)			(Mean ± SD)			* 95% CI *			* 95% CI *		
**RSI (AU)**	0.91	±	0.30	0.89	±	0.30	0.01			1.01		
							* −0.03 *	* to *	* 0.05 *	* 0.96 *	* to *	* 1.06 *
**Jump Height (m)**	0.27	±	0.05	0.26	±	0.05	0.01			0.98		
							* −0.01 *	* to *	* 0.03 *	* 0.93 *	* to *	* 1.04 *
**Flight Time (s)**	0.47	±	0.05	0.47	±	0.05	−0.01			1.03		
							* −0.02 *	* to *	* 0.00 *	* 1.00 *	* to *	* 1.05 *
**Ground Contact** **Time (s)**	0.32	±	0.11	0.32	±	0.11	0.00			1.00		
							* 0.00 *	* to *	* 0.00 *	* 1.00 *	* to *	* 1.01 *
**Mean Propulsive** **Power (W)**	2726.90	±	585.09	2678.15	±	575.27	3.03			1.02		
							* −91.00 *	* to *	* 97.06 *	* 0.98 *	* to *	* 1.05 *
**Peak Propulsive** **Power (W)**	4525.16	±	984.45	4448.20	±	961.74	−28.08			1.02		
							* −315.54 *	* to *	* 259.37 *	* 0.95 *	* to *	* 1.10 *
**Peak Propulsive** **Velocity (m/s)**	2.42	±	0.21	2.39	±	0.22	0.12			0.96		
							* −0.17 *	* to *	* 0.42 *	* 0.85 *	* to *	* 1.07 *
**Net Propulsive** **Impulse (N.s)**	195.39	±	32.36	192.60	±	31.68	−1.38			1.02		
							* −10.31 *	* to *	* 7.55 *	* 0.98 *	* to *	* 1.06 *
**Mean Propulsive** **Force (N)**	2042.37	±	425.30	2030.98	±	422.18	−3.61			1.01		
							* −22.47 *	* to *	* 15.25 *	* 1.00 *	* to *	* 1.02 *
**Peak Propulsive** **Force (N)**	3056.82	±	839.12	3047.28	±	837.84	4.92			1.00		
							* −5.46 *	* to *	* 15.29 *	* 1.00 *	* to *	* 1.01 *
**Stiffness (N/m)**	19.46	±	10.54	19.00	±	10.30	0.02			1.02		
							* −0.48 *	* to *	* 0.51 *	* 0.99 *	* to *	* 1.06 *
**Braking Depth (m)**	−0.23	±	0.07	−0.23	±	0.07	0.00			1.00		
							* −0.01 *	* to *	* 0.01 *	* 0.96 *	* to *	* 1.04 *
**Mean Braking** **Power (W)**	3550.86	±	925.72	3555.47	±	890.29	−146.10			1.04		
							* −304.22 *	* to *	* 12.02 *	* 0.99 *	* to *	* 1.09 *
**Peak Braking** **Power (W)**	8258.04	±	2570.69	8119.93	±	2228.50	−1108.73			1.15		
							* −1853.27 *	* to *	* −364.18 *	* 1.06 *	* to *	* 1.25 *
**Net Braking** **Impulse (N.s)**	224.53	±	39.35	225.68	±	38.33	−7.13			1.03		
							* −19.17 *	* to *	* 4.90 *	* 0.98 *	* to *	* 1.07 *
**Mean Braking** **Force (N)**	2611.15	±	647.27	2602.91	±	638.87	−25.97			1.01		
							* −54.40 *	* to *	* 2.46 *	* 1.00 *	* to *	* 1.03 *
**Peak Braking** **Force (N)**	4134.00	±	1163.00	4076.00	±	1106.00	−151.43			1.05		
							* −319.35 *	* to *	* 16.50 *	* 1.00 *	* to *	* 1.10 *
**Body Weight (N)**	833.00	±	142.00	833.00	±	141.00	−2.78			1.00		
							* −9.60 *	* to *	* 4.03 *	* 1.00 *	* to *	* 1.01 *

**Key:** AMTI, advanced mechanical technology, Inc.; HD, Hawkin Dynamics; SD, standard deviation; CI, confidence interval; AU, arbitrary unit; m, metres; s, seconds; N, Newtons; Red Text, indicates identified bias.

**Table 4 sensors-23-04820-t004:** Descriptive statistics for selected DJ variables.

Variables	AMTI			HD		
	(Mean ± SD)			(Mean ± SD)		
**Touch-down** **Velocity (m/s)**	−2.36	±	0.26	−2.35	±	0.25
**Fall Height (m)**	0.35	±	0.04	0.35	±	0.04

**Key:** AMTI, advanced mechanical technology, Inc.; HD, Hawkin Dynamics; SD, standard deviation; m, metres; s, seconds.

## Data Availability

Not applicable. All data (i.e., the mean and standard deviation of each metric) are shown in the tables referenced in the manuscript.

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
