# Peer review of "The Validity of Hawkin Dynamics Wireless Dual Force Plates for Measuring Countermovement Jump and Drop Jump Variablesâ€"

_sensors, 2023, doi:10.3390/s23104820_

Round 1
Reviewer 1 Report
Comments and Suggestions for Authors
Dear Authors,
the research is interesting and worth of importance.
There are methodological drawbacks you have to address (I enclose suggestions within the paper).
Best regards,

Author Response
Please find attached our responses to your comments in the attached document. Many thanks for your work in helping us to improve our manuscript.

Reviewer 2 Report
Comments and Suggestions for Authors
Reviewer’s Comments:
The manuscript “The concurrent validity of the Hawkin Dynamics wireless dual force plate system for measuring countermovement jump and drop jump force-time variables” is a very interesting work. Herein, force plate testing is becoming more commonplace in sport due the advent of commercially available, portable, and affordable force plate systems (i.e., hardware and software). Following the validation of the Hawkin Dynamics, Inc. (HD) proprietary software in recent literature, the aim of this study was to determine the concurrent validity of the HD wireless dual force plate hardware for assessing vertical jumps. During a single testing session, the HD force plates were placed directly atop two adjacent Advanced Mechanical Technology, Inc. in-ground force plates (the “gold standard”) to simultaneously collect vertical ground reaction forces produced by 20 participants during the countermovement jump (CMJ) and drop jump (DJ) tests (1000 Hz). Agreement between force plate systems was determined via ordinary least products regression using bootstrapped 95% confidence intervals. No bias was present between the two force plate systems for all CMJ and DJ variables, except DJ peak braking force (proportional bias) and DJ peak braking power (fixed and proportional bias). While I believe this topic is of great interest to our readers, I think it needs major revision before it is ready for publication. So, I recommend this manuscript for publication with major revisions.
1. In this manuscript, the authors did not explain the importance of the wireless dual force plate system in the introduction part. The authors should explain the importance of wireless dual force plate systems.
2) Title: The title of the manuscript is not impressive. It should be modified or rewritten it.
3) Correct the following statement “The HD system may be considered a valid alternative to the industry gold standard for assessing vertical jumps because no fixed or proportional bias was identified for all CMJ variables (n = 17) and only 2 out of 18 DJ variables”.
4) Keywords: The wireless dual force plate system is missing in the keywords. So, modify the keywords.
5) Introduction part is not impressive. The references cited are very old. So, Improve it with some latest literature such as 10.3389/fmats.2022.877683
6) The authors should explain the following statement with recent references, “Thus, no vGRF data from the AMTI system (which began data capture immediately before the HD system) that was recorded before touchdown was included in the analyses”.
7) Add space between magnitude and unit. For example, in synthesis “21.96g” should be 21.96 g. Make the corrections throughout the manuscript regarding values and units.
8) The author should provide reason about this statement “This is an initial concern, as it is rare that participants will perform separate CMJ trials with identical force-time characteristics”.
9. Comparison of the present results with other similar findings in the literature should be discussed in more detail. This is necessary in order to place this work together with other work in the field and to give more credibility to the present results.
10) Conclusion part is very long. Make it brief and improve by adding the results of your studies.
11) There are many grammatic mistakes. Improve the English grammar of the manuscript.
Comments on the Quality of English LanguageMinor editing of English language required
Author Response

(The authors gave the same response as above.)

Reviewer 3 Report
Comments and Suggestions for Authors
I made comments directly in the .pdf

Author Response

(The authors gave the same response as above.)

Round 2
Reviewer 1 Report
Comments and Suggestions for Authors
Dear Authors,
thanks for following my suggestions. Your reply is fine.
Best regards,